# Antibiofilm Activity and Mechanism of Linalool against Food Spoilage *Bacillus amyloliquefaciens*

**DOI:** 10.3390/ijms241310980

**Published:** 2023-07-01

**Authors:** Guanghui Shen, Lu Yang, Xinyu Lv, Yingfan Zhang, Xiaoyan Hou, Meiliang Li, Man Zhou, Le Pan, Anjun Chen, Zhiqing Zhang

**Affiliations:** 1College of Food Science, Sichuan Agricultural University, Ya’an 625014, China; shenghuishen@163.com (G.S.); m18224038942_1@163.com (X.L.); zhangyingfann@163.com (Y.Z.); houxiaoyan106@163.com (X.H.); liml@sicau.edu.cn (M.L.); zhouman@sicau.edu.cn (M.Z.); anjunc003@163.com (A.C.); 2Chemical Engineering College, Xinjiang Agricultural University, Urumqi 830052, China; inmail911@sina.com

**Keywords:** linalool, antibiofilm, *Bacillus amyloliquefaciens*, cell motility, extracellular matrix, cell surface properties, molecular docking

## Abstract

Pellicle biofilm-forming bacteria *Bacillus amyloliquefaciens* are the major spoilage microorganisms of soy products. Due to their inherent resistance to antibiotics and disinfectants, pellicle biofilms formed are difficult to eliminate and represent a threat to food safety. Here, we assessed linalool’s ability to prevent the pellicle of two spoilage *B. amyloliquefaciens* strains. The minimum biofilm inhibitory concentration (MBIC) of linalool against *B. amyloliquefaciens* DY1a and DY1b was 4 μL/mL and 8 μL/mL, respectively. The MBIC of linalool had a considerable eradication rate of 77.15% and 83.21% on the biofilm of the two strains, respectively. Scanning electron microscopy observations revealed that less wrinkly and thinner pellicle biofilms formed on a medium supplemented with 1/2 MBIC and 1/4 MBIC linalool. Also, linalool inhibited cell motility and the production of extracellular polysaccharides and proteins of the biofilm matrix. Furthermore, linalool exposure reduced the cell surface hydrophobicity, zeta potential, and cell auto-aggregation of *B. amyloliquefaciens*. Molecular docking analysis demonstrated that linalool interacted strongly with quorum-sensing ComP receptor and biofilm matrix assembly TasA through intermolecular hydrogen bonds, hydrophobic contacts, and van der Waals forces interacting with site residues. Overall, our findings suggest that linalool may be employed as a potential antibiofilm agent to control food spoilage *B. amyloliquefaciens*.

## 1. Introduction

Biofilms are cell communities of tightly associated microorganisms embedded in an extracellular matrix (ECM) of polymeric substances including exopolysaccharides (EPSs), proteins, and eDNA [1]. The ECM gathers bacterial cells together to protect them from external biological and abiotic factors. Due to the protection and buffering provided by biofilm, the resistance of bacteria embedded in the biofilm to bacteriostatic and bactericidal substances in the external environment is dozens to thousands of times higher than that of planktonic bacteria. Biofilms formed at air–liquid interfaces are termed floating biofilms or pellicles [2,3]. *Bacillus* species can not only tolerate high temperatures due to their endospores, but also readily form a complex macroscopic pellicle biofilm [4], in which bacteria embedded are equally or more resistant than those in surface-attached biofilms.

*Bacillus amyloliquefaciens* have been well studied as plant growth-promoting rhizobacteria (PGPR), which have various beneficial effects on plants’ growth, stress tolerance, and disease prevention [5]. Biofilm formation is one of the required steps of root colonization by PGPR and may help them perform their beneficial roles [6]. Although the biofilm of *Bacillus* spp. is beneficial for crop cultivation, bacteria embedded in a biofilm are also important cross-contaminating resources in the food industry. Some works have shown that *B. amyloliquefaciens* stains could cause spoilage and quality loss in raw milk [7], bread [8], and slimy rice noodles [9]. In our previous work [10], two strains of *B. amyloliquefaciens* were confirmed to be the primary spoilage bacteria in a Chinese non-fermented soy food, namely Sichuan Yuba. These two spoilers are capable of rapidly degrading soy proteins in Yuba, leading to sensory and structural quality deterioration, shortening the shelf life of the product, and forming large amounts of biogenic amines that are harmful to human health, increasing the risk to food safety. Moreover, due to the largest pellicle formation and high extracellular protease production, the presence of *B. amyloliquefaciens* pellicle leads to cross-contamination and food spoilage, thus posing a serious threat to the quality and safety of Sichuan Yuba. Therefore, finding effective treatments to combat biofilm-associated bacteria is a challenge for the Yuba food industry.

Currently, the commonly employed approaches to prevent and control bacterial biofilm in the food industry include physical, chemical, and natural agents, enzymes, and other methods, but the majority of the methods easily cause the bacteria in biofilm to develop resistance to the adverse environment [11]. Compared to conventional methods, natural compounds are promising green and safe antibiofilm agents [12]. Linalool is a floral terpene alcohol commonly found as the major volatile component in the essential oils of a variety of fragrant plants, including lavender, bergamot, and Sichuan pepper [13]. Linalool has been approved by the Joint FAO/WHO Expert Committee on Food Additives (JECFA) as a food additive with no safety concerns [14]. Numerous studies have shown that linalool and extracts containing linalool have a broad spectrum of antibacterial activity [15,16,17,18] and antibiofilm effects [19,20]. According to Lahiri et al. [21], linalool can significantly decrease the content of extracellular polysaccharides in the biofilm of *Pseudomonas aeruginosa* and had an impact on the synthesis of quorum-sensing (QS) molecules LasA and LasB. Additionally, Manoharan et al. [22] reported that linalool inhibited the biofilm of *Candida albicans* in a dose-dependent manner, with 0.005% linalool concentration reducing biofilm formation by up to 90%. Linalool has therefore been demonstrated to be an excellent phytoconstituent for microbial biofilm reduction. However, the inhibitory effectiveness of linalool against the pellicle biofilm of *B. amyloliquefaciens* and the underlying mechanisms of action have not been well elucidated. 

In the current research, we assessed the antibiofilm and eradication ability of linalool against pellicle biofilm of two spoilage *B. amyloliquefaciens* strains. Furthermore, the role of linalool in inhibiting *B. amyloliquefaciens* pellicle formation and its molecular mechanisms have been elucidated by studying cell motility, cell surface properties, and the interaction of linalool with known targets involved in biofilm formation using the molecular docking approach. This study is expected to contribute to the understanding of the intracellular antibiofilm mechanisms of linalool against *B. amyloliquefaciens* and its potential as a biofilm control agent for the soy food industry.

## 2. Results

### 2.1. The Inhibitory Effect of Linalool on B. amyloliquefaciens Pellicle

#### 2.1.1. The Minimum Biofilm Inhibitory Concentration (MBIC) of Linalool on *B. amyloliquefaciens* Pellicle 

Figure 1 depicts pellicle biofilm formation at varied linalool concentrations. From the data in Figure 1a, the MBIC values of linalool on *B. amyloliquefaciens* DY1a and DY1b were 4 μL/mL and 8 μL/mL, respectively. After 48 h of culturing, no visible pellicle biofilm was produced with a linalool concentration of 1 MBIC. In comparison to the pellicle biofilm that formed on the air–liquid surface in the absence of linalool, which was much denser and more robust and had more wrinkles, the topography of the pellicle biofilm formed at linalool concentrations of 1/4 MBIC and 1/2 MBIC appeared to be looser and more homogeneous and have a less rough surface. 

As presented in Figure 1b, two *B. amyloliquefaciens* strains had a considerable reduction in pellicle production when linalool concentration was increased. Linalool can therefore successfully prevent *B. amyloliquefaciens* from forming a pellicle at the air–liquid interface. Linalool has been shown to have effective antibiofilm action against *P. aeruginosa* [21].

#### 2.1.2. The Effect of Linalool on Viable Bacteria within Pellicle

Pellicle biofilms can gather bacterial cells together to protect them from external biotic and abiotic threats. According to Nahar et al. [23], this refuge increases bacteria’s resistance to challenging life circumstances such as dryness, nutrition restriction, and antimicrobial chemicals. Biofilm cells have a 1000-fold higher tolerance for bacteriostatic and bactericidal substances than planktonic cells [24]. Linalool significantly decreased the viable cells in the pellicle biofilm, as demonstrated in Figure 2. 

When the linalool concentration was raised from 1/4 MBIC to 1/2 MBIC, the number of viable cells decreased significantly (*p* < 0.05) compared to the control groups. As the linalool concentration increased to the MBIC, no viable cells were detected. These findings showed that linalool might be used as an antibiofilm agent for combating *B. amyloliquefaciens* biofilm formation.

#### 2.1.3. The Effect of Linalool on Microstructure Morphology of Pellicle 

The scanning electron microscope (SEM) image (Figure 3) demonstrates the inhibitory impact of linalool on the pellicle biofilm in a more obvious way. In the absence of linalool, the microstructure of the pellicle biofilm was compact and dense, and the cells embedded in the pellicle biofilm were coated with large amounts of ECM matrix, which played a critical role in supporting the biofilm’s mechanizability [25]. The number of bacterial cells decreased as linalool concentration increased, and the structure of the pellicle biofilm became more unstable and vulnerable to rupture. These findings suggested that linalool has a good controlling effect on *B. amyloliquefaciens* pellicle biofilm by interfering with the formation of a stable biofilm structure.

#### 2.1.4. The Eradication Effect of Linalool on Preformed Pellicles

As demonstrated in Figure 4, the preformed pellicle biofilm of DY1a and DY1b was reduced by 77.15% and 83.21%, respectively, after 24 h of exposure to the MBIC of linalool. These findings suggested that linalool had a concentration-dependent eradication effect on preformed mature pellicle biofilms of both strains. 

### 2.2. Effects of Linalool on the Cell Motility of B. amyloliquefaciens

Cell motility is an important feature in bacterial biofilm formation, as well as in bacterial survival and colonization [26]. The swimming circle of the control bacteria created a roughly circular colony, as shown in Figure 5, whereas the swarming circle established a particular shape due to fast radial expansion on the agar surface [27]. The diameter and size of the swimming ring created on the surface of the culture plate rapidly reduced as the concentration of linalool rose. Linalool, when added at the MBIC, utterly inhibited the flagellum-dependent swimming action of two strains (Figure 5a). Similar to the swimming results, the width and area of the bacteria’s swarming circle reduced as the concentration of linalool in the medium increased (Figure 5b). Based on these findings, we inferred that linalool’s inhibitory impact on pellicle biofilm was due to its suppression of bacterial swimming and swarming motility.

### 2.3. Effects of Linalool on the Production of Extracellular Polysaccharides and Proteins

Biofilm ECM, which is mostly made of extracellular polysaccharides, extracellular protein, and DNA (eDNA), is essential for biofilm development and architecture [28]. Extracellular polysaccharides and proteins, two main components of the biofilm matrix, were examined in this section. The contents of polysaccharides and proteins in the pellicle biofilm decreased as the concentration of linalool increased from 1/4 MBIC to 1/2 MBIC, as shown in Figure 6, and the contents of polysaccharides and proteins in the control group without linalool were significantly higher than those in the linalool supplementation group. 

Not only are extracellular polysaccharides and proteins the two primary components of the biofilm ECM, but they also play an important role in the first phase of biofilm development by aiding bacterial adherence to diverse surfaces. As a result, we investigated the levels of exopolysaccharides and proteins in the culture medium generated by the planktonic cells that did not reside in the pellicle. The contents of polysaccharides and proteins in the extracellular medium supplemented with linalool were significantly lower than those in the control group in the absence of linalool, indicating that linalool inhibited the secretion of extracellular polysaccharides and proteins by planktonic cells. 

### 2.4. Effects of Linalool on Bacterial Surface Properties of B. amyloliquefaciens

#### 2.4.1. Cell Surface Hydrophobicity

Cell surface hydrophobicity (CSH) between bacterial somatic cells is a critical physicochemical factor for the adhesion, aggregation, and proliferation of microorganisms on solid surfaces or air–liquid interfaces [29]. As shown in Figure 7a, the addition of linalool had a significant effect on cell surface hydrophobicity (*p* < 0.05). As the concentration of linalool increased from 1/4MBIC to MBIC, the cell surface hydrophobicity decreased significantly from 32.72% to 14.66% for strain DY1a and from 32.50% to 14.50% for strain DY1b. The cell surface hydrophobicity of the control group (41.74% and 40.67%) in the absence of linalool was considerably greater than that of the linalool group (*p* < 0.05).

#### 2.4.2. The Effect of Linalool on Bacterial Cell Surface Zeta Potential

Cell surface charge is a crucial factor in determining whether or not bacteria will colonize a surface [30]. As demonstrated in Figure 7b, the surface zeta potential of *B. amyloliquefaciens* strains DY1a and DY1b in the control group without linalool was −43.30 mV and −36.65 mV, respectively. The absolute value of zeta potential on the cell surface decreased significantly after linalool treatment, and as linalool concentration increased from 1/4 MBIC to MBIC, the absolute value of zeta potential on the surface of DY1a cells decreased from 37.15 mV to 29.65 mV, and the corresponding potential value of DY1b decreased from 32.10 mV to 22.80 mV. These results indicated that linalool exhibited a significant reduction in the cell surface potential of spoilage strains.

#### 2.4.3. The Effect of Linalool on Bacterial Auto-Aggregation Ability

Auto-aggregation may serve as an adhesion mechanism for the integration and establishment of bacteria in the biofilm community because it represents the contact between bacterial cells [31]. The addition of linalool reduced the rate of bacterial self-aggregation (Figure 7c), suggesting that the bacteria were more dispersed. As linalool concentration increased, auto-aggregation of DY1a and DY1b decreased to 13.84% and 18.71%, respectively. The decrease in cell surface hydrophobicity, according to thermodynamic theory, can reduce surface tension between cells while increasing tension between cells and the surrounding liquid medium [32]. The decrease in cell surface hydrophobicity reduced the energy required to uniformly disperse the cell suspension and facilitated the maintenance of the suspension, resulting in a decrease in auto-aggregation capacity.

### 2.5. Molecular Docking Analysis

To investigate the potential antibiofilm mechanism of linalool, binding interactions of linalool with the selected 67 biofilm-related receptor proteins of *B. amyloliquefaciens* were investigated by a blind docking method. The results of docking scores are listed in Appendix A (Appendix A). Among these selected receptors, linalool was predicted to interact with ComP and TasA, with the highest binding values of −6.4 and −6.3 kcal/mol, respectively. 

QS plays an essential role in regulating the formation and development of bacterial biofilms. The ComQXPA quorum sensing system, a conserved QS system in the genus *Bacillus*, is involved in a variety of physiological processes, including extracellular matrix production, which is required for the formation of pellicle biofilms [33]. ComP, a two-component sensor histidine kinase in the ComP-ComA system, acts as an essential receptor in the reception of QS signaling molecules [34]. As shown in Figure 8a, the structure of linalool complexed with ComP was used in a docking study, and the results showed that linalool effectively binds to subsite I of the ComP binding site with binding energies of −6.4 kcal/mol. Hydrogen bonds were formed between Asn179, a H-bond donor, and the oxygen atom in the hydroxyl group of linalool. Another hydrogen bond was formed between the hydrogen atom of the hydroxyl group in linalool and Ala161, a H-bond receptor. Linalool could bind to the active site of ComP via hydrophobic interactions with residues Ala307, Ala303, Leu310, Tyr162, Tyr289, and Phe285, which serve to enhance the binding of ComP. Meanwhile, a pi-sigma covalent bond was formed between linalool and Phe183, as well as van der Waals forces with residues Ala 307, Arg237, and Val 168 in the docking pocket, allowing the linalool-ComP complex to remain stable. 

Bacterial cells in the pellicle biofilm are surrounded by ECM, which consists mainly of extracellular polysaccharides, proteins such as amyloid-like fibers (ALFs), and extracellular nucleic acids. TasA, as one of the matrix’s major protein components, is a precursor of ALF in *Bacillus* and is important in the early stages of pellicle biofilm development [35]. TasA can also form fibers, which are an important component of the pellicle biofilm matrix [36]. Bohning et al. [37] recently confirmed TasA’s critical role in scaffolding *B. subtilis* biofilms by forming sheet-rich fibers and the resulting bundles from monomers. Figure 8b demonstrates that linalool may bind to the active site of TasA via hydrophobic interactions with Tyr181, Ile173, Lys186, and Ala1. Furthermore, van der Waals forces formed with residues Ile80, Asn74, Thr187, and Val184 of TasA can help the linalool-TasA complex stay stable. According to these findings, linalool has a high affinity for the QS histidine kinase ComP and matrix assembly TasA proteins. According to the report by Verma et al. [38], the FDA-approved small molecule inhibitors lovastatin and simvastatin completely inhibited biofilm formation and had an effective disruptive impact on preformed *B. subtilis* biofilms. Further molecular docking and dynamics simulation analysis revealed that lovastatin formed stable interactions with TasA. Therefore, it can be inferred that targeting vital components of the extracellular matrix, especially interfering with the assembly of TasA fibers, may be a potential mode of action for linalool as an antibiofilm agent. The docking-based simulation approach can be used as a reliable tool for identifying novel molecular targets and elucidating the mode of action of antibiofilm candidates.

## 3. Discussion

*Bacillus* spp. are common harmful bacteria in the food processing environment; they can not only form endospores with strong stress resistance under appropriate environmental conditions [39], but also form biofilms that are difficult to be removed, further enhancing the adhesion ability of bacteria and spores in food environment [40]. Due to these characteristics of *Bacillus*, the production and distribution enterprises experience huge economic losses, not only affecting the nutritional quality and shelf life of products, but also directly threatening the health of consumers, causing serious food safety problems. *B. amyloliquefaciens* can rapidly form pellicle biofilms at the air–liquid interface [41] and also produce non-pathogenic spores, enhance drug resistance, and easily cause food spoilage [7,8,9]. In this study, linalool at 4 μL/mL and 8 μL/mL could completely inhibit the pellicle biofilm formation of *B. amyloliquefaciens* DY1a and DY1b, respectively. Therefore, the use of linalool is an effective strategy to combat pellicle biofilm formation of spoilage *B. amyloliquefaciens*. 

Current biofilm control strategies are divided into three categories: changing abiotic surface qualities to avoid biofilm development, regulating signaling pathways to inhibit biofilm formation and promote biofilm dispersal, and using external pressures to remove biofilms [42]. Preventing the development of biofilm formation is more effective and precise than the destruction of the mature biofilm; thus, the use of combinatorial therapy with significant efficacy rather than mono-therapeutic antibiotic treatment was proposed [25]. Natural or synthetic antibiofilm agents have different modes of action against various bacteria to inhibit biofilm development, such as membrane permeabilization, QS signaling blocking, peptidoglycan cleavage, and inhibition of bacterial cell division [25]. The analysis of the regulatory signal pathway involved in biofilm formation is an indispensable step in the study of biofilm inhibition mechanisms. The current techniques used to decipher the mechanism of antibiofilm agents include metabolomics, proteomics, and transcriptomics. However, these omics methods are resource-intensive, making them ineffective for drug discovery and mechanism analysis. Molecular docking is a computer-based virtual approach used to analyze a chemical substance’s binding affinity to a target protein to further sequence potential targets. The approach proposed has offered an economical approach for identifying protein targets for large-scale small-molecule sets, with the advantages of closely correlating with phenotypic change, lowering probing costs, and enhancing target prediction accuracy [43]. Recently, a few studies have been conducted to screen drugs that have inhibitory effects on biofilms through molecular docking. Nosran et al. [44] observed a strong significant affinity between the pyochelin-zingerone conjugate and FptA, pyochelin’s outer membrane receptor, using molecular docking methods. An in vitro experiment demonstrated that this conjugate suppressed swimming, swarming, and twitching motilities as well as biofilm formation. Zayed et al. [45] synthesized substituted fluoroquinazolinones with evident antibiofilm activity through chemical synthesis, and further revealed that these derived compounds had strong binding with their receptor sites through molecular docking, and detected their expected binding mode. 

In the present study, molecular docking results indicated that linalool could bind to the histidine kinase ComP and TasA proteins, resulting in an indirect suppression of bacteria motility and interfering with the production of extracellular polysaccharides and amyloid fibrin, thus inhibiting the pellicle growth of *B. amyloliquefaciens*. Therefore, molecular docking-based simulation can be a reliable tool for the rapid identification of novel molecular targets and elucidation of the mode of action of potential natural antibiofilm agents. Although we identified the ComP and TasA as two essential receptors for linalool among 67 pellicle biofilm formation-related receptors of *B. amyloliquefaciens* by molecular docking, it is necessary to further validate the potential regulatory mechanism of linalool against pellicles by real-time expression changes of relevant downstream genes and in vitro binding assays. 

Bacterial biofilm formation is a complex dynamic process, which includes the initial adhesion and colonization, EPS secretion, ECM scaffold, and microcolony formation [42]. In *B. subtilis*, for building pellicle biofilms at air–water interfaces, cells switch from a planktonic to a sessile state by upregulating the expression of genes involved in the production of the extracellular matrix. This switch is led by the reinforcement of external cues, for example, nutrient depletion, low oxygen levels, and surface adherence [46]. In the early stages of pellicle formation, Steinberg et al. [47] observed a significant number of cell aggregates in the center of the air–liquid interface and concluded that flagella-dependent clustering vortex motion might expedite pellicle biofilm growth. Therefore, the irreversible adhesion of planktonic cells to the pellicle biofilm-forming interface through cluster aggregation plays an important role in pellicle formation at the early stage. This behavior is affected by cell chemotactic movement, cell–cell adhesion, and the synthesis of ECM components including extracellular polysaccharides and scaffolding proteins. Also, Van der Waals forces and the hydrophobic properties of both the abiotic surfaces and bacterial cells are also essential factors influencing the cell adhesion. Linalool significantly suppressed cell swimming and swarming, delaying the formation of pellicle biofilm, according to cell motility data. Additionally, linalool also decreased cell surface hydrophobicity and cell surface charge, on which bacterial adhesion to surfaces greatly depends. 

The hydrophobicity and charge of the cell surface are connected to the adhesin protein and lipopolysaccharide on the bacterial cell surface. Under normal physiological conditions, molecules with electronegative groups, such as lipopolysaccharides and glycoproteins, are distributed on the surface of bacterial cells and exhibit strong electronegativity [48]. The antibiofilm activity by biosurfactants and EPS produced by bacteria has been attributed to the fact that groups such as hydroxyl and amino groups in these molecules could alter cell hydrophobicity and cell surface charges [49]. The free hydroxyl groups in linalool molecules could attract positively charged groups and produce rearrangement [50] during interaction with the bacterial surface components, thus resulting in decreased cell surface electronegativity and hydrophobicity of *B. amyloliquefaciens*. Linalool could act similarly to biosurfactants reported as able to modify the charge and hydrophobicity of the cell surface, hence affecting cell-to-cell aggregation. This mechanism of biofilm inhibition is similar to the mode of action of rhamnolipid surfactants [51]. Linalool also inhibited the synthesis and secretion of polysaccharides and proteins, thereby affecting the initial adhesion process of bacteria, resulting ultimately in the delay or inhibition of mature pellicle formation. 

## 4. Materials and Methods

### 4.1. Bacterial Strains and Culture

The two strains of *B. amyloliquefaciens*, DY1a and DY1b, were isolated from contaminated Sichuan Yuba by our group [10]. Both strains were incubated in Luria–Bertani (LB) Broth at 37 °C for 24 h, shaking at 160 rpm. For pellicle biofilm development, both strains were incubated in modified Tryptic Soy Broth supplemented with 5% soymilk power (TSBS). Linalool with 98% purity was purchased from Shanghai Rhawn Chemical Technology Co., Ltd. (Shanghai, China).

### 4.2. Antibiofilm Activity Assessment

#### 4.2.1. Minimum Biofilm Inhibitory Concentration (MBIC) Determination

The two-fold serial broth microdilution assay [52] was used to determine MBIC values. Briefly, 5 mL bacteria solution with 10^6^ CFU/mL was transferred to a 50 mL beaker containing 5 mL sterile TSBS. Afterward, 500 μL/mL linalool dissolved in ethanol was added individually to each beaker to obtain final linalool concentrations ranging from 1.0 to 8.0 µL/mL. A culture medium replaced with 10.0 μL/mL absolute ethanol was used as the negative control. After incubation at 37 °C for 48 h, the minimum concentration with no visible growth of pellicle biofilm compared with the control group was recorded as the MBIC. The pellicle biofilm of different treatment groups was gently harvested and then weighed. 

#### 4.2.2. Enumeration of Viable Bacteria in Biofilm

Pellicle biofilms formed at the air–liquid interface of TSBS were prepared according to the slide attachment method [53] with slight modification. A disinfected glass slide (76 mm × 26 mm) was placed vertically and partially immersed in each 50 mL beaker containing 20 mL TSBS medium supplemented with different concentrations of linalool (1/4 MBIC, 1/2 MBIC, and MBIC) and cultured at 37 °C for 48 h. Ten μL/mL absolute ethanol was used as the control. For enumeration of viable cells present in the pellicle biofilm, a specific area of pellicle biofilm (19.7 mm × 25.6 mm) attached to the slide was collected and gently rinsed with sterile PBS (pH 7.2) and then subjected to sonication for 30 min at 300 W power, followed by vortex mixing for 10 s to disperse the pellicle biofilm. After that, the suspensions containing detached bacteria were serially diluted 10-fold, and then 50 μL appropriate dilutions were spread on PCA plates and cultured at 37 °C for 48 h; the number of viable cells was measured using the colony counting methodology [54]. 

#### 4.2.3. Scanning Electron Microscope (SEM)

The formed pellicle biofilms attached on slides cultured using the glass slide attachment method as described above were rinsed with sterile PBS (pH 7.2) to remove the nonadherent bacterial cells and then transferred onto a sterile coverslip (1.0 cm × 1.0 cm) and fixed with 2.5% glutaraldehyde for 2 h. Samples were then rinsed with distilled water and dehydrated in graded ethanol (30%, 60%, 80%, 90%, and 100%) for 15 min each. Finally, the pellicle biofilm architecture was then observed using an SEM (ZEISS EVO 18, Carl-Zeiss, Oberkochen, Germany) after gold spraying. Cultures without linalool served as the control. 

#### 4.2.4. Biofilm Eradication Effect 

The eradication effect of linalool on preformed mature pellicles was measured using the 2,3,5-triphenyl tetrazolium chloride (TTC) reduction assay [55]. TTC is utilized to distinguish metabolically active cells from inactive ones. Various intracellular dehydrogenases in living bacterial cells enzymatically reduce TTC to red 1,3,5-triphenylformazan (TFP). Spectrophotometry is used to estimate the rate of reduction, which is utilized as an indirect measure of bacterial metabolic activity [56]. A volume of 0.5 mL bacterial suspension (10^6^ CFU/mL) was transferred to a 24-well plate with 1.5 mL TSBS and incubated at 37 °C for 48 h. The preformed pellicle in each well was rinsed twice with sterile PBS and transferred to 2 mL of linalool solution with final concentrations of 1/4 MBIC, 1/2 MBIC, and MBIC. The pellicle exposed to linalool was then incubated at 37 °C for 24 h. To measure the metabolic activity of the pellicle biofilm, the linalool solution in the 24-well plate was discarded, and 1.5 mL of 1.0 mg/mL TTC solution was then added. After reaction at 37 °C for 3 h in the dark, 200 μL of the solution was transferred to a new 96-well plate, and the final absorbance was read at 490 nm using a microplate reader (Varioskan Flash, Thermo Fisher Scientific, Waltham, MA, USA). The pellicle biofilm formed in wells without linalool but containing 10.0 μL/mL ethanol was used as the control. The eradication effect of linalool on the preformed pellicle was expressed as a reduction in biofilm metabolic activity.

### 4.3. Cell Motility Assays

Cell motility was measured according to the method described by Zhang et al. [57]. For the swimming assay, 5 μL of 10^6^ CFU/mL bacterial suspension was inoculated with sterile needles on the center of a swimming plate (peptone 10.0, NaCl 5.0, glucose 5.0, and agar 4.0 g/L, pH 7.0) containing linalool with final concentrations of 1/4 MBIC, 1/2 MBIC, and MBIC. For swarming motility, the purified single colony of spores was dipped with sterile toothpicks and pierced into the swarming plates (peptone 10.0, NaCl 5.0, glucose 5.0, and agar 5.0 g/L, pH 7.0) but not pierced to the bottom of the plate. After inoculation, the culture plates were incubated at 37 °C for 24 h, and the diameter of the bacterial migration or turbidity zone was recorded. Plates without linalool but containing 10.0 μL/mL ethanol were used as a control.

### 4.4. Measurement of Pellicle ECM Components

Pellicle biofilms were cultured and harvested according to the method described in Section 4.2.1. Briefly, EPS in the pellicle ECM was extracted using the method of Rajitha et al. [58]. The collected pellicle biofilms were transferred to a 10 mL centrifuge tube, and 5 mL of 2% EDTA solution (containing 2% NaCl) was added. After extraction with shaking at 37 °C for 12 h, the pellicle biofilm suspensions were centrifuged at 4,000 r/min for 30 min, and the resulting EPS-extracted supernatant was collected for determination of the content of polysaccharides and proteins. Furthermore, 2.0 mL of bacterial suspension under the floating biofilm was pipetted and centrifuged at 10,000 r/min for 10 min, and the supernatant was subjected to further analysis of EPS components in cell suspensions. The polysaccharide content of EPS was measured using a previously modified phenol–sulfuric acid method, and the protein content of EPS was determined using the Coomassie brilliant blue method [57]. 

### 4.5. Effects of Linalool on Bacterial Cell Surface Properties 

#### 4.5.1. Cell Surface Hydrophobicity

Bacterial cell surface hydrophobicity was assessed by microbial adherence to n-hexadecane in the microbial adhesion to hydrocarbons (MATH) assay in accordance with Zoueki et al. [59] with some modifications. Briefly, cells were harvested by centrifugation at 4000 r/min for 5 min, rinsed three times with sterile PBS, and finally resuspended in sterile PBS to achieve an OD600 of 0.5. Then, 3.0 mL of bacterial suspension was added to 9 mL of linalool solution with different final concentrations of 1/4 MBIC, 1/2 MBIC, and MBIC. The mixture was then incubated for 4 h at 37 °C. A layer of 0.15 mL of *n*-hexadecane and 1.5 mL of xylene was placed on top of the bacterial solution. The phases were separated by leaving them to remain at room temperature for 15 min after vortexing for 120 s. At 600 nm, the absorbances of the original mixture (*A*_0_) and the aqueous phase (*A*) were measured. The cell surface hydrophobicity rate was obtained using the following equation:(1)$$\mathrm{Cell}\;\mathrm{surface}\;\mathrm{hydrophobicity}\;\mathrm{rate}\;(\%)\;=\frac{A_{0}-A}{A_{0}}\text{×}100\;$$

#### 4.5.2. Zeta Potential

The zeta potential of bacterial cells exposed to different concentrations of linalool was measured by referring to the method of de Aguiar et al. [60]. In brief, 5 mL of linalool solution was added to 5 mL of bacterial culture with an OD600 of 0.20 to achieve final concentrations of 1/4 MBIC, 1/2 MBIC, and MBIC and then transferred to a 50 mL centrifuge tube and incubated for 2 h at 37 °C with shaking at 160 r/min. The zeta potential of the bacterial suspension was assessed using a Zetasizer Nano ZS (Malvern Instruments Ltd., Worcestershire, UK) after the suspension was dispersed by ultrasound for 10 min. 

#### 4.5.3. Cell Auto-Aggregation 

The auto-aggregation of bacterial cells exposed to linalool was assessed using the method of Wang et al. [61]. Briefly, 3 mL of bacterial suspension with OD600 of 0.25 was introduced to a 15 mL test tube containing 9 mL of various doses of linalool solution (1/4 MBIC, 1/2 MBIC, and MBIC). After vortex mixing for 10 s, 200 µL of the bacterial solution was transferred to a 96-well plate and incubated at 37 °C for 6 h. A microplate reader was used to measure the absorbance of the upper layer before incubation (*A*_1_) and after incubation (*A*_2_) at 600 nm. The auto-aggregation (%) was obtained using the following formula:(2)$$\mathrm{Auto}-\mathrm{aggregation}\;\mathrm{rate}\;(\%)\;=\frac{A_{1}-A_{2}}{A_{2}}\text{×}100\;$$

### 4.6. Molecular Docking of Linalool with Potential Receptors

The 3D structure of linalool was acquired from the PubChem database available at http://pubchem.ncbi.nlm.nih.gov/ (accessed on 21 December 2022) and then optimized with the MMFF94 force field using Avogadro software (Version 1.2.0) [62]. For collecting potential biofilm-related targets in *B. amyloliquefaciens*, we comprehensively searched the relevant literature in the Web of Science, Scopus, and PubMed databases using the keywords “Biofilm” and “*Bacillus*”, and then the obtained targets from the studies found were intersected with the whole genes of *B. amyloliquefaciens* type strain DSM7 to obtain the intersected genes as the target receptors for molecular docking study. The Protein Data Bank (PDB) database does not contain the experimentally established 3D structures of the matched targets involved in biofilm development. As a result, the full-length amino acid sequences of 67 potential protein targets were obtained from the National Center for Biotechnology Information (NCBI) genome database (https://www.ncbi.nlm.nih.gov/nuccore/FN597644.1) (accessed on 23 September 2022) and submitted to the Robetta online server from Baker Lab of University of Washington (https://robetta.bakerlab.org/) [63] to predict 3D models of the proteins using the homology modeling approach. The best predictive models of each target were selected as receptors for further docking simulations. The CB-DOCK2 online server from Yang Cao Lab of Sichuan University (http://cao.labshare.cn/) [64] was used to perform cavity prediction and blind docking of linalool with each potential target receptor. The docking scores were calculated using the ΔG binding energy values (Kcal/mol). The best-docked complexes were downloaded, and their 3D interaction visualization was performed using Discovery Studio 2021.

### 4.7. Statistical Analysis

All experiments were carried out in triplicates. The experimental results were presented as means ± standard deviations and analyzed using SPSS 22.0 for one-way ANOVA with Dunnett’s multiple comparison tests at a significance level of *p* < 0.05.

## 5. Conclusions

In summary, our study revealed that linalool exerted good antibiofilm as well as pellicle biofilm eradication activity against spoilage *B. amyloliquefaciens* of Sichuan Yuba. Linalool inhibited pellicle biofilm formation by interfering with the cell motility related to adhesion and aggregation in the early stage of pellicle biofilm formation, decreasing cell surface hydrophobicity and zeta potential, as well as inhibiting extracellular polysaccharide and protein components of the ECM. Molecular docking confirmed the interactions of linalool with the ComP receptor and TasA fibers, two essential targets involved in the pellicle formation of *B. amyloliquefaciens*. The present investigation suggests that linalool could be applied as a potential antibiofilm agent for pellicle biofilm control of spoilage *B. amyloliquefaciens* in the soy food industry. 

## Figures and Tables

**Figure 1 ijms-24-10980-f001:**
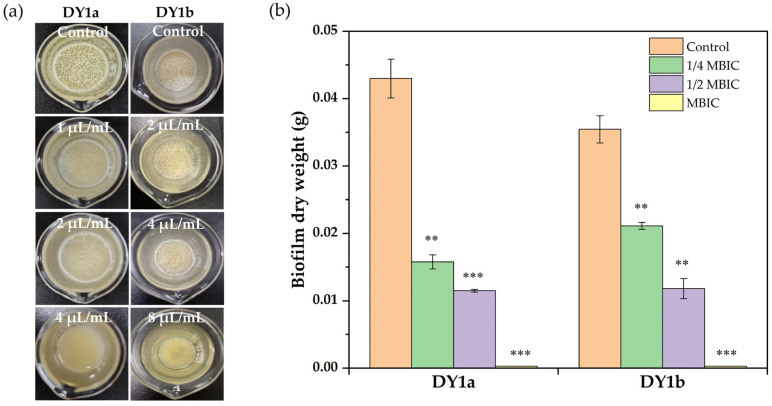
Effects of linalool on pellicle biofilm formation (**a**) and yield (**b**) of *B. amyloliquefaciens* DY1a and DY1b. The lowercase letters indicate that there were significant differences between the control and experimental groups. One-way ANOVA followed by Dunnett’s multiple comparison post hoc test was carried out to determine statistical significance between each treatment and the negative control of the same strain; ** *p* < 0.01, *** *p* < 0.001.

**Figure 2 ijms-24-10980-f002:**
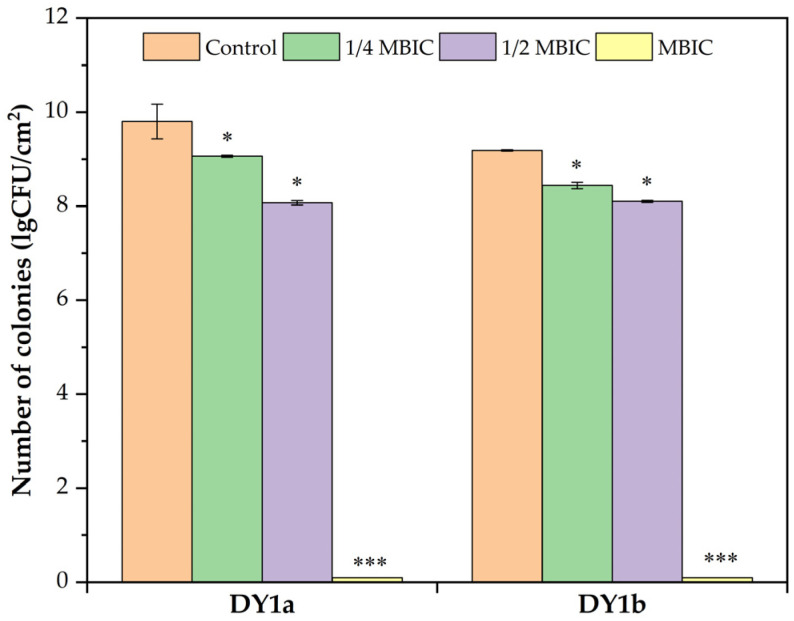
Effects of linalool on viable counts of living bacteria in the pellicle of *B. amyloliquefaciens* DY1a and DY1b. One-way ANOVA followed by Dunnett’s multiple comparison post hoc test was carried out to determine statistical significance between each treatment and the negative control of the same strain; * *p* < 0.05, *** *p* < 0.001.

**Figure 3 ijms-24-10980-f003:**
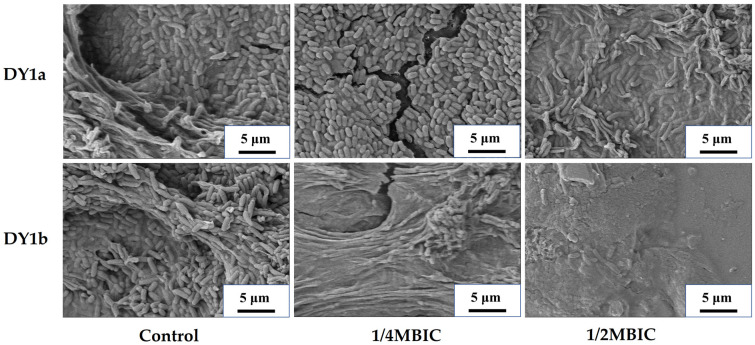
Effects of linalool on pellicle microstructure of *B. amyloliquefaciens* DY1a and DY1b.

**Figure 4 ijms-24-10980-f004:**
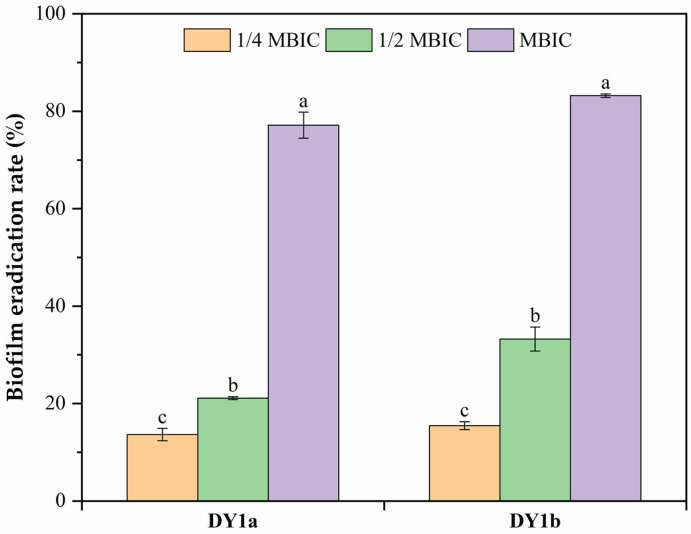
Eradication effects of linalool on preformed pellicle biofilm of *B. amyloliquefaciens* DY1a and DY1b. One-way ANOVA followed by Dunnett’s multiple comparison post hoc test was carried out to determine statistical significance among treatments of the same strain; different lowercase letters above columns of the same strain indicate differences at *p* < 0.05.

**Figure 5 ijms-24-10980-f005:**
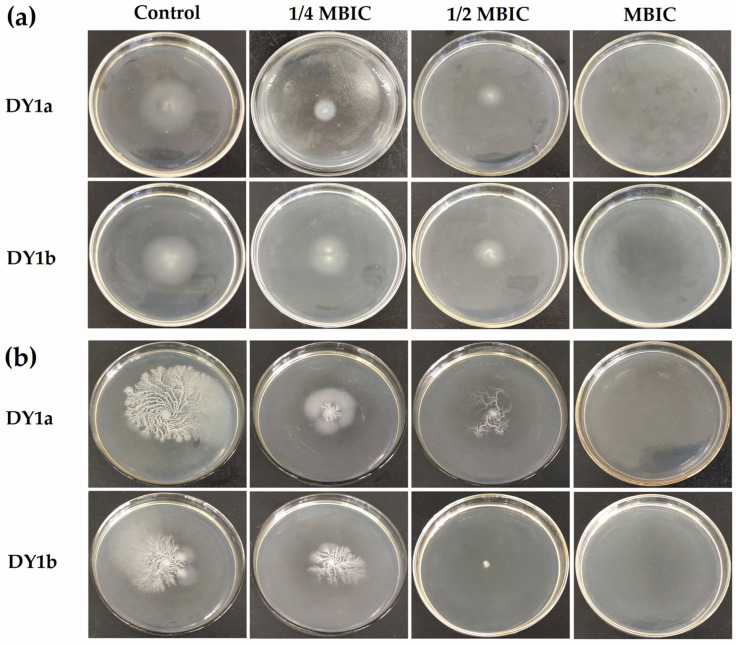
Effects of linalool on swimming (**a**) and swarming motility (**b**) of *B. amyloliquefaciens* DY1a and DY1b.

**Figure 6 ijms-24-10980-f006:**
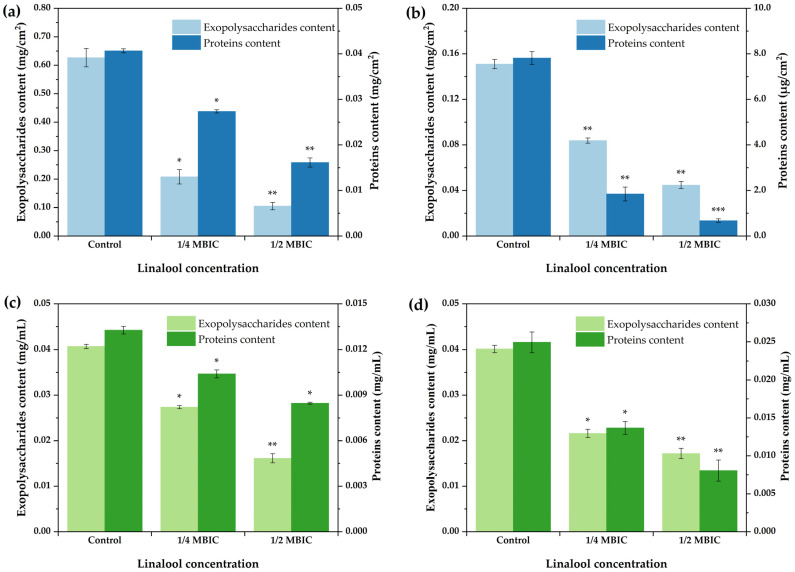
Effects of linalool on exopolysaccharide and protein contents in pellicle matrix (**a**,**b**) and culture medium (**c**,**d**). (**a**,**c**) DY1a; (**b**,**d**) DY1b. One-way ANOVA followed by Dunnett’s multiple comparison post hoc test was carried out to determine statistical significance between each treatment and the negative control of the same strain; * *p* < 0.05, ** *p* < 0.01, *** *p* < 0.01.

**Figure 7 ijms-24-10980-f007:**
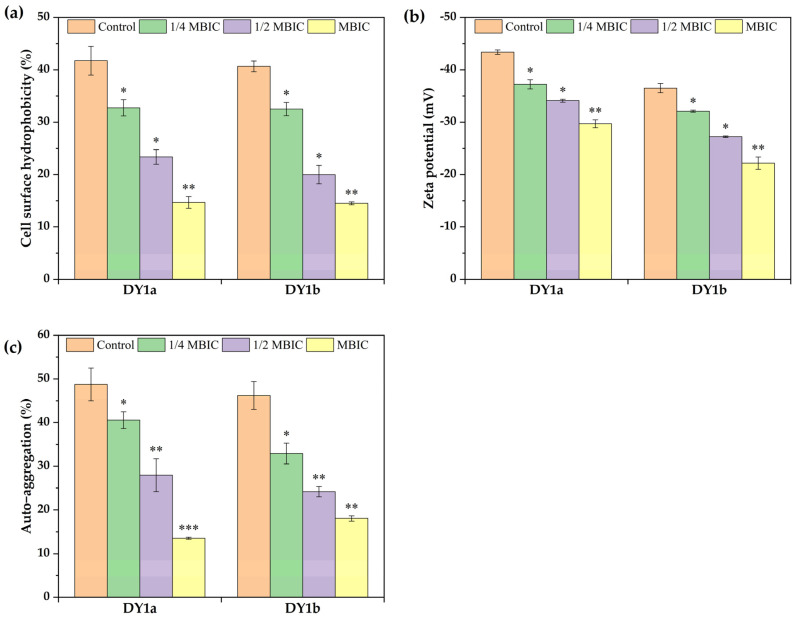
Effects of linalool on cell surface hydrophobicity (**a**), cell surface Zeta potential (**b**), and auto-aggregation ability (**c**) of *B. amyloliquefaciens* DY1a and DY1b. One-way ANOVA followed by Dunnett’s multiple comparison post hoc test was carried out to determine statistical significance between each treatment and the negative control of the same strain; * *p* < 0.05, ** *p* < 0.01, *** *p* < 0.001.

**Figure 8 ijms-24-10980-f008:**
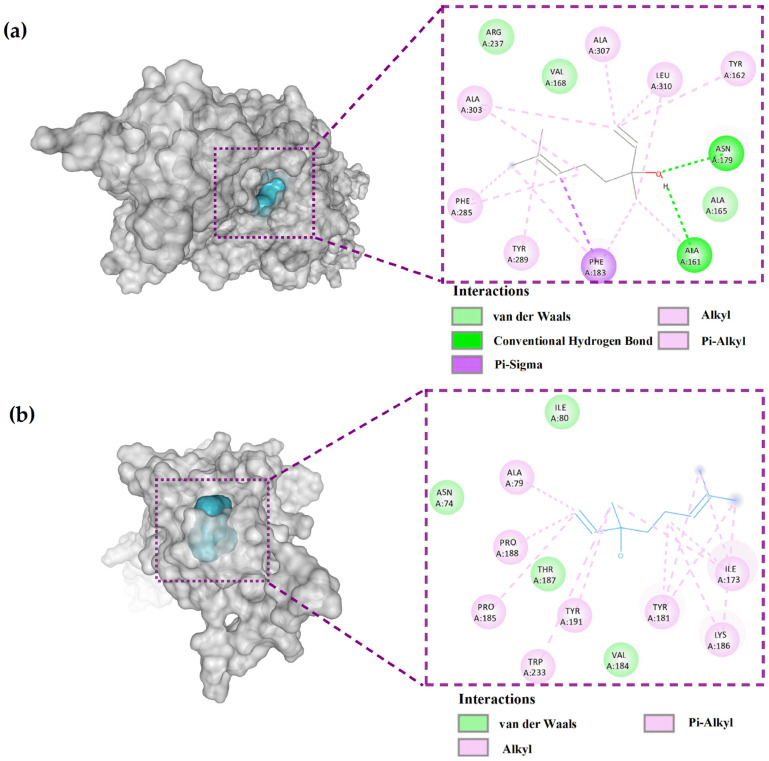
The 3D (left) and 2D (right) interaction of linalool with ComP (**a**) and TasA (**b**) of *B. amyloliquefaciens*.

## Data Availability

Data will be made available on request.

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
