# Peer review of "Antibiofilm Activity and Mechanism of Linalool against Food Spoilage *Bacillus amyloliquefaciens"

_ijms, 2023, doi:10.3390/ijms241310980_

Round 1

Reviewer 1 Report

In the paper submitted for review, Authors described the effect of linalool on B. amyloliquefaciens biofilm formation. Authors used many analysis to check the influence the linaloon on biofilm like zeta potential, SEM pictures, cell surface hydrofobicity, cell auto-aggregation etc. I appreciate the using molecular docknig analysis. These analyzes are not often found in biofilm research, so it is interesting to the reader. Therefore, I recomend accept this paper after minor revision.

"When the linalool concentration was raised from 1/4 MBIC to 1/2 MBIC, the number of viable cells decreased significantly (P < 0.05)." - was a siginiticant difference between control and 1/4MBIC value? The reader does not necessarily see which differences are statistically significant, and the description does not contain complete information. You might want to think about adding some additional information in the description or on the chart.

Please, make sure that all Latin names of the strain are in italics and remove double spaces in the tekst.

Author Response

Dear editors and reviewers,

Thank you very much for your comments concerning our manuscript entitled “Antibiofilm activity and mechanism of linalool against food spoilage Bacillus amyloliquefaciens” (ijms-2444871). The comments were helpful for revising and improving our paper. We have studied the comments carefully and we made appropriate revisions. The main corrections are marked in red in this revised manuscript. Below please find our point-by-point responses to the reviewers’ comments.

If there are any other modifications we could make, we would like very much to modify them and we really appreciate your help.

Kind regards!

First author: Guanghui Shen

E-mail: shenghuishen@163.com

Corresponding author: Prof. Zhiqing Zhang

E-mail: zqzhang721@163.com.

College of Food Science, Sichuan Agricultural University

Response to Reviewer 1

Comment 1: "When the linalool concentration was raised from 1/4 MBIC to 1/2 MBIC, the number of viable cells decreased significantly (P < 0.05) compared with the control." - was a significant difference between control and 1/4MBIC value? The reader does not necessarily see which differences are statistically significant, and the description does not contain complete information. You might want to think about adding some additional information in the description or on the chart.

Response 2: We thank the reviewer for raising this point regarding the data statistic of viable cells number. We are sorry for the ambiguity caused by the expression. As pointed out also by the reviewer, the description in the submitted manuscript did not contain complete information. We have rephrased this sentence to increase clarity, and provided a more detailed description of statistical symbols in Figure 1.

Comment 2: Please, make sure that all Latin names of the strain are in italics and remove double spaces in the tekst.

Response 2: We thank the reviewer for this thoughtful comment, all the Latin names of the strain has been amended to italics at lines 116, 201, 270 in the revised manuscript.

Reviewer 2 Report

This research paper describes the antibiofilm effect of linalool on Bacillus amyloliquefaciense involved in food spoilage. The authors investigated the antibiofilm mechanisms of linalool using motility assay, extracellular polysaccharides and proteins assays, cell surface hydrophobicity, zeta potential, auto-aggregation as well as molecular docking. These approaches appear to be fine to analyze the antibiofilm properties to the molecular levels. The manuscript is very well-written and organized. I have very few additional comments as below.

1. There are some misspellings. Line 78, 84: B. amylolifaciens B. amyloliquefaciens.

2. Figure 1(b): In statistical analysis, ‘letter grouping’ was used and each sample is labeled with ‘a’, ‘b’, ‘c’, or ‘d’. It would be better to indicate which samples were compared. It looks like that both DY1a and DY1b were analyzed separately. For example, in control and 1/4MBIC, both DY1a and DY1b are labeled the same letter although the values are apparently different. So, I would say ‘Each sample was statistically analyzed separately’ or ‘Different letters in the same strain are significantly different’ in the figure caption. This is same for Fig 2, 4, and 6.

3. Line 95: ‘such dryness’ ‘such as dryness’

4. In Figure 2, ‘Contorl’ ‘Control’

5. Line 196-198: Please make sure that this is a correct information. To me, the decrease in cell surface hydrophobicity would precipitate the cells, inhibit the maintenance of the suspension, and increase the auto-aggregation capacity.

6. Some of the reference numbers do not match very well in the text. Line 234: Reference [44] in the reference list is ‘Verma et al.’. In the same way, in line 240-241, Reference [45] in the reference list is ‘Andre et al.’. In line 254-255, the reference for ‘dairy products’ appears to be reference [47], not [48] and the reference for ‘bread’ appears to be reference [48], not [49].  

Author Response

Dear editors and reviewers,

Thank you very much for your comments concerning our manuscript entitled “Antibiofilm activity and mechanism of linalool against food spoilage Bacillus amyloliquefaciens” (ijms-2444871). The comments were helpful for revising and improving our paper. We have studied the comments carefully and we made appropriate revisions. The main corrections are marked in red in this revised manuscript. Below please find our point-by-point responses to the reviewers’ comments.

If there are any other modifications we could make, we would like very much to modify them and we really appreciate your help.

Kind regards!

First author: Guanghui Shen

E-mail: shenghuishen@163.com

Corresponding author: Prof. Zhiqing Zhang

E-mail: zqzhang721@163.com.

College of Food Science, Sichuan Agricultural University

Response to Reviewer 2

Comment 1: There are some misspellings. Line 78, 84: B. amylolifaciens → B. amyloliquefaciens.

Response 1: We are sorry for our carelessness, this typo has been corrected (Line 92,99).

Comment 2: Figure 1(b): In statistical analysis, ‘letter grouping’ was used and each sample is labeled with ‘a’, ‘b’, ‘c’, or ‘d’. It would be better to indicate which samples were compared. It looks like that both DY1a and DY1b were analyzed separately. For example, in control and 1/4MBIC, both DY1a and DY1b are labeled the same letter although the values are apparently different. So, I would say ‘Each sample was statistically analyzed separately’ or ‘Different letters in the same strain are significantly different’ in the figure caption. This is same for Fig 2, 4, and 6.

Response 2: We are sorry for the ambiguity caused by the expression, we have redrawn Figure 1, 2, 4, 6, 7, more clear presentation regarding statistics has been added in each caption of figures.

Comment 3: Line 95: ‘such dryness’ → ‘such as dryness’

Response 3: We have corrected it at Line 112 in the revised manuscript.

Comment 4: In Figure 2, ‘Contorl’ → ‘Control’

Response 4: We have corrected it in the revised Figure 2.

Comment 5: Line 196-198: Please make sure that this is a correct information. To me, the decrease in cell surface hydrophobicity would precipitate the cells, inhibit the maintenance of the suspension, and increase the auto-aggregation capacity.

Response 5: We sincerely appreciate the valuable comments. The results of these experimental measurements mentioned by the reviewer have been repeatedly verified by our group. We can guarantee the accuracy and authenticity of the experimental results. In addition, by searching the literature, results of some relevant studies are consistent with results presented in this manuscript, eg: (1) https://www.mdpi.com/2304-8158/11/15/2230,(2) https://doi.org/10.1016/j.foodcont.2020.107667, (3) https://doi.org/10.4014/jmb.2109.09045. (4) https://doi.org/10.3390/foods11152230. The different results between the above mentioned studies and other studies with opposite correlation available to the reviewer may be attributed to the cell surface characteristics of the strains used, growth stages, cell hydrophobicity test conditions, and culture medium. These interesting differences deserve further investigation and discussion.

Comment 6: Some of the reference numbers do not match very well in the text. Line 234: Reference [44] in the reference list is ‘Verma et al.’. In the same way, in line 240-241, Reference [45] in the reference list is ‘Andre et al.’. In line 254-255, the reference for ‘dairy products’ appears to be reference [47], not [48] and the reference for ‘bread’ appears to be reference [48], not [49]. 

Response 6: Thanks for your careful checks. We are sorry for our carelessness. Based on your comments, we have checked and corrected all of the cited references in the order of their appearance within the whole manuscript.

Reviewer 3 Report

The article " Antibiofilm activity and mechanism of linalool against food spoilage Bacillus amyloliquefaciens" describe the effect of linalool on two strains of B. amyloliquefaciens

Before accepting this article for publication, some modifications need to be performed:

- in the introduction some references are missing, for examples : doi: 10.3389/fmicb.2021.562094. eCollection 2021 and doi: 10.1016/j.micpath.2020.103980. Epub 2020 Jan 19

- it is not clear how different are the strains DY1a and DY1b, also in the cited article this aspect is not clear. The behaviour of the two are so similar that I am not sure they can be defined as two different strains, Were they sequenced? maybe it would be better to add another type of Bacillus to increase the significance of the results

- in all the experiments a positive control using a known antimicrobial is missing

-how common is the spoilage by B. amyloliquefaciens? What are its main negative effects?

- how were the 67 proteins for the docking study  selected? This information is missing and it should be added in results and in materials and methods

- figure legends should all be improved, especially letters related to statistics should be defined

- line 96 and line 35, which tolerance?

- comment better the influence of Z-potential and attachment, add  references to other publications

-Bacillus amyloliquefaciens has positive effects in soil, it is considered  a PGPM, please add this clarification in introduction

-please define MBIC and others

-title contain an error : remove last "e" in amyloliquefaciense

Only monor errors are present, for example sometimes B. amyloliquefaciens is not in italic

Author Response

Dear editors and reviewers,

Thank you very much for your comments concerning our manuscript entitled “Antibiofilm activity and mechanism of linalool against food spoilage Bacillus amyloliquefaciens” (ijms-2444871). The comments were helpful for revising and improving our paper. We have studied the comments carefully and we made appropriate revisions. The main corrections are marked in red in this revised manuscript. Below please find our point-by-point responses to the reviewers’ comments.

If there are any other modifications we could make, we would like very much to modify them and we really appreciate your help.

Kind regards!

First author: Guanghui Shen

E-mail: shenghuishen@163.com

Corresponding author: Prof. Zhiqing Zhang

E-mail: zqzhang721@163.com.

College of Food Science, Sichuan Agricultural University

Response to Reviewer 3

Comment 1: in the introduction some references are missing, for examples : doi: 10.3389/fmicb.2021.562094. eCollection 2021 and doi:10.1016/j.micpath.2020.103980. Epub 2020 Jan 19

Response 2: We thank the reviewer for identifying these missing references and we have inserted them where indicated (now corresponding to lines 69, 572-573 and 578-579).

Comment 2: it is not clear how different are the strains DY1a and DY1b, also in the cited article this aspect is not clear. The behaviour of the two are so similar that I am not sure they can be defined as two different strains, Were they sequenced? maybe it would be better to add another type of Bacillus to increase the significance of the results

Response 2: We thank the reviewer for this thoughtful comment. In our previous study, more than ten strains responsible for spoilage of Yuba have been isolated, including these two B. amyloliquefaciens DY1a and DY1b, Bacillus subtilis DY2a, DY3, etc.(https://www.spkx.net.cn/fileup/1002-6630/HTML/2018-39-2-028.shtml). These strains’ spoilage capacity, and biofilm formation capacity study found that DY1a and DY1b have a greater ability to cause spoilage and biofilm formation than other strains. Therefore, we subsequently focused our research efforts on the biofilm control of these two strains. Although both DY1a and DY1b were identified as B. amyloliquefaciens by 16s rDNA and gyrB gene sequencing analysis, we found that strain DY1b had a higher capacity of biofilm formation, and more resistant to food preservatives compared to DY1a (http://sf1970.cnif.cn/CN/10.13995/j.cnki.11-1802/ts.018641;

https://xuebao.scau.edu.cn/zr/hnny_zren/article/abstract/20200413?st=search). Indeed, in this research, the MBIC value of linanlool against DY1b was 8 μL/mL, which is twice as high as that of DY1a. Therefore, the results of this study have important guiding meaning for the control of B. amyloliquefaciens in Yuba industry. Inspired by the reviewers’ valuable comments, more strains of Bacillus genus isolated from different foods would be investigated in our further study to increase the significance of linalool’ antibiofilm ability.

Comment 3: in all the experiments a positive control using a known antimicrobial is missing

Response 3: Thanks for reviewer’s helpful suggestions. Our this research mainly focused on determining linalool’ antibiofilm activity against spoilage B. amyloliquefaciens and exploring the underlying molecular mechanism. Inspired by the valuable comments of the reviewers, we will further consider the selection of different known antimicrobial for comparative studies in the next step of performing application validation tests in Yuba product production and storage.

Comment 4: how common is the spoilage by B. amyloliquefaciens? What are its main negative effects?

Response 4: we are greatly appreciate the comments. In fact, B. amyloliquefaciens strains have general spoilage-causing effects on products such as dairy products, rice and pasta products, and bakery breads. We have added some relevant literatures at lines 46-51 in the revised manuscript.

Comment 5: how were the 67 proteins for the docking study selected? This information is missing and it should be added in results and in materials and methods.

Response 5: Thank you for your suggestions. We are sorry for missing of this key methodological information. We have added the information regarding these 67 targets collection and screening methods in lines 492-497.

Comment 6: figure legends should all be improved, especially letters related to statistics should be defined

Response 6: Thank you for your valuable suggestions. All the figure legends have now been redrawn on what we feel is an easier to understand. letters represented statistics results have defined in figures captions.

Comment 7: line 96 and line 35, which tolerance?

Response 7: We are sorry for these ambiguous expressions. We have revised these two sentences, and clarified the subject of resistance or tolerance.

Comment 8: comment better the influence of Z-potential and attachment, add references to other publications

Response 8: We appreciate the reviewer’s careful reading of the manuscript. We have added two references and further discuss the mode of action of Z-potential and attachment on biofilm formation, at lines 362-364, 368-371 in the revised manuscript.

Comment 9: Bacillus amyloliquefaciens has positive effects in soil, it is considered a PGPM, please add this clarification in introduction

Response 9: Thanks for the reviewer for this suggestion. We have added text to describe Bacillus amyloliquefaciens as plant growth-promoting rhizobacteria (PGPR), as well as the role of biofilm formation in their root colonization and beneficial functions (lines 41-44 in revised manuscript).

Comment 10: please define MBIC and others

Response 10: We have defined MBIC at line 89, SEM at line 126, EPS at line 31, NCBI at line 500.

Comment 11: title contain an error : remove last "e" in amyloliquefaciense

Response 11: We are sorry for our carelessness, this typo has been corrected.

Reviewer 4 Report

The study conducted by Shen et al. has scientific value, is well designed and conducted, with original and relevant contributions in biofilm control of spoilage Bacillus amyloliquefaciens in soy food industry. I support its publication after appropriate minor modifications as outlined below.

Title: Bacillus amyloliquefaciense is wrong. Please correct it!

Introduction

Line 56, line 59, lines 87-88, line 94, etc.: In case of the authors cited in the article (e.g., Lahiri et al.; Manoharan et al.) you shouldn't put a comma between the first author's name and et al. Please correct this.

Results

The subchapters are numbered incorrectly. Please correct this!

Lines 78; 84; 100: you should write B. amyloliquefaciens instead B. amylolifaciens. Please correct it!

Line 100, Figure 2: it is a drafting error, you should write control instead of contorl. Please correct it.

Lines 118-121: This paragraph is appropriate for the "Materials and Methods" section and not for the "Results" section. Please transfer it.

Line 180: The species name must be italicized. Please correct it!

Discussion

The authors must extend biofilm control strategies discussion. In this regard I suggest the authors to improve the reference list with valuable recently published article about the mechanisms of action of antibiofilm agents. (https://www.ncbi.nlm.nih.gov/pmc/articles/PMC9394423/)

Materials and Methods

The subchapters are numbered incorrectly. Please correct this!

Line 332: Minimum biofilm inhibitory concentration (MBIC) determination: please insert the appropriate reference.

References

In the list of references, the authors cited in the text are not listed in the order of their appearance in the manuscript, as required by the journal. Please check and correct!

Author Response

Dear editors and reviewers,

Thank you very much for your comments concerning our manuscript entitled “Antibiofilm activity and mechanism of linalool against food spoilage Bacillus amyloliquefaciens” (ijms-2444871). The comments were helpful for revising and improving our paper. We have studied the comments carefully and we made appropriate revisions. The main corrections are marked in red in this revised manuscript. Below please find our point-by-point responses to the reviewers’ comments.

If there are any other modifications we could make, we would like very much to modify them and we really appreciate your help.

Kind regards!

First author: Guanghui Shen

E-mail: shenghuishen@163.com

Corresponding author: Prof. Zhiqing Zhang

E-mail: zqzhang721@163.com.

College of Food Science, Sichuan Agricultural University

Response to Reviewer 4#

Comment 1: Title: Bacillus amyloliquefaciense is wrong. Please correct it!

Response 1: We are sorry for our carelessness, this typo has been corrected.

Comment 2: Introduction Line 56, line 59, lines 87-88, line 94, etc.: In case of the authors cited in the article (e.g., Lahiri et al.; Manoharan et al.) you shouldn't put a comma between the first author's name and et al. Please correct this.

Response 2: We apologize for this non-conforming literature citation format. All of the authors cited have been corrected, at lines 70, 72, 111, 269, 276, 319, 323, 345, 435, 448, 462 in the revised manuscript.

Comment 3: The subchapters are numbered incorrectly. Please correct this!

Response 3: We are sorry for our carelessness, the subchapters numbers have been corrected.

Comment 4: Lines 78; 84; 100: you should write B. amyloliquefaciens instead B. amylolifaciens. Please correct it!

Response 4: We are sorry for our carelessness, the used strains’ Latin name have been revised.

Comment 5: Line 100, Figure 2: it is a drafting error, you should write control instead of contorl. Please correct it.

Response 5: We have corrected it in revised Figure 2.

Comment 6: Lines 118-121: This paragraph is appropriate for the "Materials and Methods" section and not for the "Results" section. Please transfer it.

Response 6: Thank you for your valuable suggestions. The mentioned paragraph has been transferred to “Materials and methods” section, lines 417-421 in the revised manuscript.

Comment 7: Line 180: The species name must be italicized. Please correct it!

Response 7: We have italicized all the Latin name of bacteria used.

Comment 8: The authors must extend biofilm control strategies discussion. In this regard I suggest the authors to improve the reference list with valuable recently published article about the mechanisms of action of antibiofilm agents. (https://www.ncbi.nlm.nih.gov/pmc/articles/PMC9394423/)

Response 8: As suggested by the reviewer, we have read this reference recommended by the reviewer and extended the biofilm control strategies in Discussion section, lines 303-308 in the revised manuscript.

Comment 9: The subchapters are numbered incorrectly. Please correct this!

Response 9: We are sorry for our carelessness, this typo has been corrected.

Comment 10: Line 332: Minimum biofilm inhibitory concentration (MBIC) determination: please insert the appropriate reference.

Response 10: We have added an appropriate reference at line 384 in the revised manuscript.   

Comment 11: In the list of references, the authors cited in the text are not listed in the order of their appearance in the manuscript, as required by the journal. Please check and correct!

Response 11: We have added some important references, and cited and listed all of references in the order of their appearance in the revised manuscript.

Round 2

Reviewer 3 Report

Authors have considered all the suggestions. Manuscript is now ok for publication